# Large-deviations of disease spreading dynamics with vaccination

**Yannick Feld** [ID]*, **Alexander K. Hartmann** [ID]

Institut für Physik, Carl von Ossietzky Universität Oldenburg, Oldenburg, Germany

* yannick.feld@uni-oldenburg.de

## Abstract

We numerically simulated the spread of disease for a Susceptible-Infected-Recovered (SIR) model on contact networks drawn from a small-world ensemble. We investigated the impact of two types of vaccination strategies, namely random vaccination and high-degree heuristics, on the probability density function (pdf) of the cumulative number $C$ of infected people over a large range of its support. To obtain the pdf even in the range of probabilities as small as $10^{-80}$, we applied a large-deviation approach, in particular the $1/t$ Wang-Landau algorithm. To study the size-dependence of the pdfs within the framework of large-deviation theory, we analyzed the empirical rate function. To find out how typical as well as extreme mild or extreme severe infection courses arise, we investigated the structures of the time series conditioned to the observed values of $C$.

**Data Availability Statement:** All relevant data are within the paper and its Supporting information files.

**Funding:** The simulations were performed at the HPC Cluster CARL, located at the University of Oldenburg (Germany) and funded by the DFG

## Introduction

Due to the high relevance for the societies, the studying of the spread of diseases has long since become a very important problem in a variety of disciplines like biology, applied mathematics, statistics and statistical physics [1–5]. Beyond the analysis of the dynamics of existing diseases one main target is to understand how to fight them. One of the best ways to mitigate the impact of a disease is the application of vaccines, which have been successfully used to eradicate smallpox [6]. For other diseases like rabies [7] and measles [8], vaccines have also led to tremendous success, although this success is undermined by anti-vaccination movements [9–12]. The creation and evaluation of new vaccines is a delicate, complex and time consuming process [13–15]. In particular, the safety and effectiveness of new vaccines has to be proven rigorously [16, 17].

To model the effect of vaccinations one has to first model the disease itself. A very famous and influential model, the Susceptible-Infected-Recovered (SIR) model, was introduced by Kermack and McKendrick in 1927 [18] who build upon previous research by Ross and Hudson [19–23]. Initially the model was investigated for a fully-mixed population [24, 25]. This means, the disease can propagate in between all individuals with the same likelihood. In physics this corresponds to a mean-field model. Naturally those early studies ignored the effect of heterogeneity within the population and thus later studies [26, 27] have incorporated contact networks, where each node represents, e.g., an individual and the edges represent the contacts between them.

through its Major Research Instrumentation Program (INST 184/157-1 FUGG) and the Ministry of Science and Culture (MWK) of the Lower Saxony State. This work also used the Scientific Compute Cluster at GWDG, the joint data center of Max Planck Society for the Advancement of Science (MPG) and University of Göttingen. Y. Feld has been financially supported by the German Academic Scholarship Foundation (Studienstiftung des Deutschen Volkes). The funders had no role in study design, data collection and analysis, decision to publish, or preparation of the manuscript.

**Competing interests:** The authors have declared that no competing interests exist.

Note that the SIR process can be understood as special case of chemical reaction networks, where the fractions of susceptible, infected and recovered correspond to the concentrations of chemical species [28, 29]. In these networks the nodes typically denote the species while the edges describe the chemical reactions. Although diffusion of species is often included in the networks, this and all other reactions are usually described on mean-field level, similar to what was done for the first SIR models.

In its simplest form, vaccination can be modeled as granting perfect immunity to the disease, i.e., an individual that has been vaccinated cannot contract the disease forevermore. Here we consider the case that a vaccination is applied before the first infected individual is introduced in the network. This could be the case for a population of domestic animals which is partially vaccinated before the disease is first introduced to the population via a wild animal. This approach is equivalent to studying the disease dynamics on a new network, where all nodes that could successfully be vaccinated have been removed from the original network. Thus, the disease can only propagate within connected components [30] of the new network and one can model the vaccination as site percolation problem [31] and usually aims at reducing the network below its percolation threshold [30, 32].

This viewpoint means that one has to achieve a rather high fraction of vaccinated individuals and ignores the effects of stochastic fluctuations that can play an important role in disease extinction [33]. Therefore it makes sense to model the stochastic process that governs the disease propagation if we want to fully understand the impact of vaccines.

To gain an understanding of the impact a disease has before going extinct one can, e.g., look at the probability density function $P(C)$ for the cumulative number $C$ of infections, i.e., how many nodes contracted the disease over its lifespan. In order to obtain a comprehensive description of such stochastic processes, one should obtain the desired distribution over a large range of its support. For few very simple models this can be done analytically, but for most interesting cases one has to apply numerical simulations [34]. In order to access a distribution even in the *large-deviation* tails which exhibit probabilities as small as $10^{-50}$ or lower one has to go beyond typical-event simulation and has to use special large-deviation techniques [35], like the ones the authors of the present work have applied in a previous study [36] where we investigated the pure SIR model comprehensively. In particular we found that the *large-deviation principle* [37–39] is fulfilled, which indicates that the model belongs to a standard class in large-deviation theory. Furthermore, our approach allowed us to obtain correlation patterns with other measurable quantities, which further deepens the understanding of the disease model and of different phases which are visible in the distribution $P(C)$.

In the present work we study the impact of two types of widely investigated vaccination strategies, i.e., random vaccination and target vaccination where nodes with a high-degree are vaccinated first [40]. For an overview of different vaccination strategies and the state of research in general we refer to detailed review articles [41, 42]. In our work we aim at obtaining $P(C)$ over a large range of the support, identify different phases in the distribution, relate these phases to different patterns of disease evolution and verify whether the mathematical large-deviation principle still holds.

The paper is structured as follows. First we introduce the SIR model and the network ensemble. Then we introduce the algorithms we applied, which includes the vaccination strategies. Next we briefly investigate the available parameter space to choose which parameters we use for the later analysis. We present the large-deviation analysis where we display the calculated probability density functions as well as corresponding rate functions and correlations with other quantities. We finish the paper with a summary and an outlook.

## Model

The SIR model we apply is defined as follows. For a given network, each node is in either of four states *Susceptible* (S), *Infected* (I), *Recovered* (R) or *Vaccinated* (V), where the vaccinated state is equivalent to the recovered state and the distinction is just made for our convenience.

To begin a simulation we assign all nodes the state S and then first apply one of the vaccination strategies that will be later discussed in Sec Vaccination strategies to assign a subset of nodes the vaccinated state V.

Then, different from the case without vaccinations [36], we choose 5 initial infected nodes, randomly and uniformly drawn from all possible nodes without allowing duplicates. Thus, the not so relevant case that just by chance the initial infection dies out quickly is much less common. This means, the trivial peak of $P(C)$ at $C \approx 0$ is much less prominent in our results. Note that vaccinated nodes can also be chosen among the initial infected one. In this case initial infection takes prevalence for the initialization, to obtain actually 5 infected nodes. However, for the later times of the disease dynamics, vaccinated nodes can never be infected, as desired.

The actual disease dynamics is performed as follows: At each time step $\tau$ we iterate over all susceptible neighbors of all currently infected nodes. Note that this implies susceptible nodes with multiple infected neighbors will be iterated over multiple times. With the *transmission probability* $\lambda$ we flag them to become infected. Next we iterate over all currently infected nodes again, this time we let each of them recover with the *recovery probability* $\mu$. To finalize the time step we infect all nodes that were flagged to become infected.

At time step $\tau$ we denote the current fraction of infected and recovered nodes by $i(\tau)$ and $r(\tau)$, respectively. From those fractions we can obtain the cumulative fraction of infected nodes, i.e., $c(\tau) = i(\tau) + r(\tau)$.

This can also be used to create an indicator for the severity of a disease outbreak by defining

$$C \equiv \lim_{\tau \to \infty} c(\tau) = \lim_{\tau \to \infty} r(\tau), \tag{1}$$

where the latter equality obviously only holds for $\mu > 0$.

## Network ensemble

We chose a *small-world network* [43] to represent our contact network. This is motivated by the fact that many real-world contact-networks exhibit small-world properties [44].

The creation of a network with $N$ nodes works as follows. First a initial ring structure is created where each node $i$ is connected to its next and second next neighbors given by $i + 1$ mod $N$ and $i + 2$ mod $N$, respectively. Since all edges are undirected this means each node starts with an initial degree of 4.

Next, we iterate over all edges $\{i, j\}$ and rewire them with probability $p$ to a random node $j' \neq i$, i.e., we exchange $\{i, j\} \to \{i, j'\}$. Throughout this paper we use a rewiring probability of $p = 0.1$. Note that we only draw a random node for $j$ because each edge is "rooted" at one node and thus the minimal degree a node exhibits is 2. This rewiring is responsible for the small-world properties of the network as it introduces shortcuts and leads to a rather small *diameter* [30], i.e., from each node one can reach any other node via a short path of edges.

## Algorithms

### Large deviation algorithms

We apply special *large-deviation* algorithms [45]. Such algorithms exist since the 1950s [46]. In statistical physics their popularity first increased for studying the dynamics of molecules via

*transition path sampling* [47, 48]. Since then, such algorithms were applied for many different models, e.g., the resilience of power grids [49, 50], random walks [51–54], random graph properties [55–58], longest increasing subsequences [59, 60], ground states of Ising spin glasses [61], and the Kardar-Parisi-Zhang equation [62].

The general idea for the SIR model is that the large-deviation simulation embraces the SIR simulation [36], i.e., it is able to manipulate the SIR dynamics in a controlled fashion to get access to the desired properties.

This manipulation works as follows: In a standard SIR simulation one would draw random numbers that are uniformly distributed in [0, 1] on demand. These numbers are then used for comparisons with the recovery probability $\mu$ or the transmission probability $\lambda$ to decide whether a node becomes recovered or infected. Instead of creating the random numbers on demand one could, of course, draw them beforehand and store them in the two vectors $\xi_\mu$ and $\xi_\lambda$, such that these vectors contain a distinct random number for each node and each time step $\tau$.

As long as the random number vectors contain enough numbers such that the disease outbreak dies out before running out of numbers, this in and of itself will not change the outcome. We refer to [36] for a detailed explanation of the estimation of the required vector length. Note that the general idea remains the same as in the referenced work, but we have slightly modified the criterion by increasing the threshold and factor compared to our previous study.

Before the SIR simulation is started the vaccination is applied, which essentially equals the removal of the vaccinated nodes. The specifics of how the nodes for the vaccination are chosen depends on the applied heuristics. All heuristics, however, require a vector $\xi_{\text{ord}}$ to give a deterministic outcome, as explained in Sec Vaccination strategies. This vector is created by shuffling the vector $[0, 1, \ldots, N - 1]$ of node IDs.

Lastly we also need to decide which nodes will be set to the infected state before the SIR simulation is started, i.e., which nodes should be "patient zero". For this we create the vector $\xi_0$ by shuffling a new vector initialized with $[0, 1, \ldots, N - 1]$. In this work we always start with 5 initial infections. Thus the first five entries of the vector determine the initially infected nodes.

To be able to sample rare events we now control the values contained within the vectors ($\xi_\mu$, $\xi_\lambda$, $\xi_{\text{ord}}$, $\xi_0$) with a Markov-Chain-Monte-Carlo (MCMC) approach that is sitting on top of the SIR simulation, i.e., the outcome of the SIR simulation now depends deterministically on the random numbers encountered during the MCMC. This MCMC is set up in a way to allow the estimation of a large range of the probability-density function (pdf) $P(C)$ for a given network. To be more precise we apply the $1/t$ *Wang-Landau* (WL) algorithm [63], which is a slight modification of the original *Wang-Landau* algorithm [64] that prevents error saturation [63, 65–67].

Since the approach we use here is very similar to the one we used for [36] we will only outline the general idea. The WL approach starts with a non-normalized estimate for the probability distribution, i.e., $P(C) = 1 \, \forall C$. Each step $t = 0, 1, \ldots$ of the Markov-Chain consists of the generation of a new *trial configuration* $\Xi' = (\xi'_\mu, \xi'_\lambda, \xi'_{\text{ord}}, \xi'_0)$, which is generated from the current configuration $\Xi^{(t)} = (\xi_\mu^{(t)}, \xi_\lambda^{(t)}, \xi_{\text{ord}}^{(t)}, \xi_0^{(t)})$ as explained in Sec MCMC Moves. Since both configurations of random vectors deterministically determine the outbreak courses they correspond to, they also determine the resulting cumulative fractions $C'$ and $C^{(t)}$ of infections, which are calculated by simulating a complete run of the SIR simulation. The trial configuration is accepted, i.e., $\Xi^{(t+1)} = \Xi'$ and thus $C^{(t+1)} = C'$, with the Metropolis-Hastings probability $\min\{1, P(C^{(t)})/P(C')\}$. If the trial configuration is not accepted we keep the old configuration, i.e., $\Xi^{(t+1)} = \Xi^{(t)}$.

A multiplicative factor $f > 1$ is used to update the pdf estimate at $C^{(t+1)}$, i.e., $P(C^{(t+1)}) = fP(C^{(t+1)})$. All other values of $P(C)$ are not changed for the current step. Due to this the

probability of generating the trial configuration with value $C$ becomes, in the long run, inversely proportional to the probability $P(C)$ of occurrence. This allows for a rather uniform sampling of the space of $C$ values while the pdf estimate is continuously refined. Usually a rather large factor $f = e \approx 2.71$ is used initially, which forces a quick but rough estimate of $P(C)$. Then some schedule is used to iteratively reduce $f$ during the process to change the estimate on an increasingly finer scale. Here lies the main difference between the $1/t$ WL algorithm we applied [63] and the original WL algorithm [64].

This method is able to measure very rare events that are inaccessible via typical-event sampling methods, also called simple sampling, and thereby allows for the sampling of distinct features of the pdf over a very large range or even the full support.

Since $P(C)$ is updated continuously, the WL method does not strictly fulfill *detailed balance* [68]. To ensure a correct statistics in the end, we subsequently employ *entropic sampling* [69]. This approach uses the pdf estimated by WL as a starting point and works as the WL approach, but $P(C)$ is not updated any more. The obtained histogram for the encountered values of $C$ is used to finally correct the pdf obtained by WL. This slightly refines the accuracy of our measurements even in the region of very low probabilities like $10^{-100}$. But more than that it enables us to more evenly sample disease infection time series over the range of $C$ values. Overall this rigorous numerical approach ensures high confidence in the results and has proven very successful in the past.

## MCMC moves

Here we will explain in detail how the trial configuration is created.

With a probability of 1% a *rotation move* is performed. That means that the vectors $\xi_\mu$ and $\xi_\lambda$ are rotated by $N$ elements to the right (50%) or left (otherwise), which roughly corresponds to shifting the resulting time series by one time step to the left or right. Note that this can be implemented quite efficiently by only storing a offset variable and thus removing the need to actually copy a lot of memory for the operation.

With a probability of 0.5% we perform a *exchange patient move*, which is done by drawing $a \in \{0, \ldots, 4\}$ and $b \in \{5, \ldots, N-1\}$ and then swapping the values $\xi_0[a] \leftrightarrow \xi_0[b]$, which effectively exchanges one of the initial infected nodes.

With a probability of 2% we perform a *walk patient move* that was found to improve convergence. The basic idea of this move is letting one randomly chosen patient of the initial patients perform one step of a random walk on the underlying (not vaccinated) network structure while making sure that each node has the same probability of occurring. This is ergodic with respect to the positions of the initial patients because we demand a connected network for this study in the first place.

For this move, we first draw one of the initial patients $p_0 \in \{\xi_0[0], \ldots, \xi_0[4]\}$. Then we draw a uniformly distributed random number $u \in [0, 1]$. Let $A$ be the adjacency list of $p_0$, i.e., a list that contains every node $p_0$ is connected to via an edge. Let $d_{\max}$ be the maximal degree of the network. Then the new patient zero $p'_0$ becomes

$$p'_0 = \begin{cases} A[i] & \text{if } \dfrac{i}{d_{\max}} \leq u < \dfrac{i+1}{d_{\max}} \\ p_0 & \text{otherwise} \end{cases} \qquad (2)$$

and $\xi_0$ is changed accordingly. Note that the finite probability of nothing changing is necessary to ensure detailed balance: The naive approach of always moving the patient would bias the system in such a way that nodes with higher degree would have a higher probability of being chosen.

With a probability of 1% we do a *reset start move*, i.e., we redraw all random numbers from $\xi_\mu, \xi_\lambda$ that are associated with the time step $\tau = 0$. This move was included since our testing suggested that it improves the convergence of the algorithm.

Lastly, if *none of the above moves* was selected, which happens with a probability of 95.5%, we just randomize the dynamics a bit. Thus, we repeat the following 3000 times: Draw a uniformly distributed random number $u \in [0, 1]$, draw one of the vectors $\xi \in \{\xi_\mu, \xi_\lambda\}$ and uniformly choose a random index $i$ of the vector to exchange the random number of the entry, i.e., $\xi[i] = u$. The number 3000 of times was chosen to achieve an acceptance rate of roughly 50% of this move.

## Histogram interval width

Since we are vaccinating at the beginning of the SIR simulation and our vaccination assumes an effectiveness of 100% this means that we have to estimate how many nodes the disease can still reach. It is, e.g., possible that some nodes are surrounded by vaccinated nodes and are therefore unreachable by the disease, if the disease does not start on them. From an algorithmic-technical point of view, we need to determine a maximum for $C$ because Wang-Landau will not work if a bin of the histogram is impossible to reach.

So we initialize the system and employ a greedy heuristics that only uses part of the Markov chain. For each step of the Markov chain we calculate the largest connected component that contains an initial infection by first removing all vaccinated nodes except those chosen as initial patients and then using a depth first search starting with the patients zero. Then we do a Markov move, where we only allow moves that change $\xi_{\text{ord}}$ or $\xi_0$ and calculate again. We reject the move if the size of the largest connected component decreased.

After 50000 steps we take the maximal size of the largest connected component that we encountered as estimate. If we later notice that the large deviation algorithm is unable to reach the largest $C$ values we remove the bins that the algorithm was not able to reach and rerun the simulation, which happened only a few times.

Note that nodes that are vaccinated can still be chosen as patients zero and in that case are not removed from the network because we consider that the vaccination was performed too late and was therefore ineffective.

## Vaccination strategies

We denote the total number of vaccination doses with $N_v$ and its fraction by $n_v = N_v/N$, where $N$ is the total number of nodes, as usual.

We consider a total of three vaccination strategies:

1. A uniformly random vaccination heuristics. It is the simplest vaccination strategy [40] that does not assume any knowledge of the topology whatsoever. For this we randomly and uniformly draw $N_v$ nodes, while not allowing duplicates. As explained in Sec Large deviation algorithms, the large-deviation scheme will supply the vector $\xi_{\text{ord}}$ as randomness input for the vaccination schemes. Conveniently we can obtain the indices of the nodes we want to vaccinate by just taking the first $N_v$ entries of $\xi_{\text{ord}}$.

2. A natural choice for a targeted vaccination strategy is to mainly target highly connected nodes [32] which is the main idea of the high degree heuristics. For this we take the $N_v$ nodes with the highest degrees possible.
   Note that the choice of the $N_v$ nodes is rarely unique as usually there are many nodes with the same degree. Thus, the choice of the small-degree vaccinated nodes must be done by some algorithm. In this case we randomly draw from the list of nodes with the

corresponding degree. This can be accomplished by selecting nodes of the same degree in an order as given by the vector $\xi_{\text{ord}}$, starting with the nodes of highest degree. Note that for this heuristics to be efficient, the degree distribution is obviously not allowed to be homogeneous. This vaccination scheme has been proven to be especially effective for scale-free networks [70].

3. The adaptive high degree heuristics is taking the idea one step further. This time we first take the node with the highest degree (if this is not unique we randomly and uniformly choose one of the possible nodes by using the $\xi_{\text{ord}}$ vector). Then, and this is the only difference to the previous strategy, we reduce the degree of all of the neighbors of the node that is now set to be vaccinated. This process is repeated until $N_v$ nodes are chosen. In the literature this heuristic is also called recalculated degree removal [71].

## Simple sampling

Before performing the relatively expensive large-deviation simulations we wanted to determine suitable values of the parameters to investigate. We chose to keep $\mu = 0.14$ from our previous study [36] and chose a rather large value of $\lambda = 0.4$ for the transmission probability, which corresponds to a parameter-space location within the epidemic phase if it was without vaccinations.

For each network size we want to study later we randomly drew one network, which means that we were very likely to obtain a typical network of our ensemble, and measured within simple-sample simulations the average $\bar{C}$ and its variance $\sigma^2(C)$ for different fractions $n_v$ of vaccinated nodes. The results for $N = 3200$ are shown in Fig 1 as example.

As expected the random vaccination strategy gives the worst results and requires the largest fraction of vaccinated nodes to achieve a certain value of $\bar{C}$. The adaptive high degree heuristics is just slightly better than the not-adaptive one only in a very small window around $n_v \approx 0.1$. On the other hand the non-adaptive one is better in the interval $n_v \in [0.1, 0.23]$. There is basically no difference between the two anywhere else. Below we discuss how the counter-intuitive behavior that the adaptive vaccination strategy in some cases performs worse can be understood.

But first we discuss how we obtained critical threshold values $n_v^c$. We fitted Gaussians (not shown) to the peak regions of the variance curves to find the exact position of the peak, which we tread as network-specific critical vaccination values $n_v^c$. In Fig 1 these critical values are visualized by the dashed lines. Note that the worse performance of the adaptive strategy on the interval [0.1, 0.23] results in a higher value of $n_v^c$ for this strategy.

We have performed this type of simulation and analysis for all networks that we want to study with the large-deviation algorithm. Since we always used the exact same networks when comparing different strategies, we use only one network for each simple-sampling study per system size $N$ that is considered later on.

Note that within our preliminary simulations we also tried to average over multiple networks for each considered size $N$ and then use finite-size scaling to find the critical vaccination doses for an infinite system, similar to what we did to obtain $\lambda_c$ in [36]. We found out that the actual distributions such as $P(C)$ still depend somehow on the actual network, especially for the high degree heuristics. Therefore, the infinite-size critical vaccination does not help much in selecting suitable values of $n_v$. Thus, we opted to use network-specific values of $n_v^c$, which fluctuate slightly: We obtained critical values for $N \in [1414, 6400]$ and found $n_v^c \in [0.351, 0.364]$ for the random heuristics, $n_v^c \in [0.166, 0.186]$ for the adaptive high-degree heuristics and $n_v^c \in [0.132, 0.150]$ for the non-adaptive high degree heuristics.

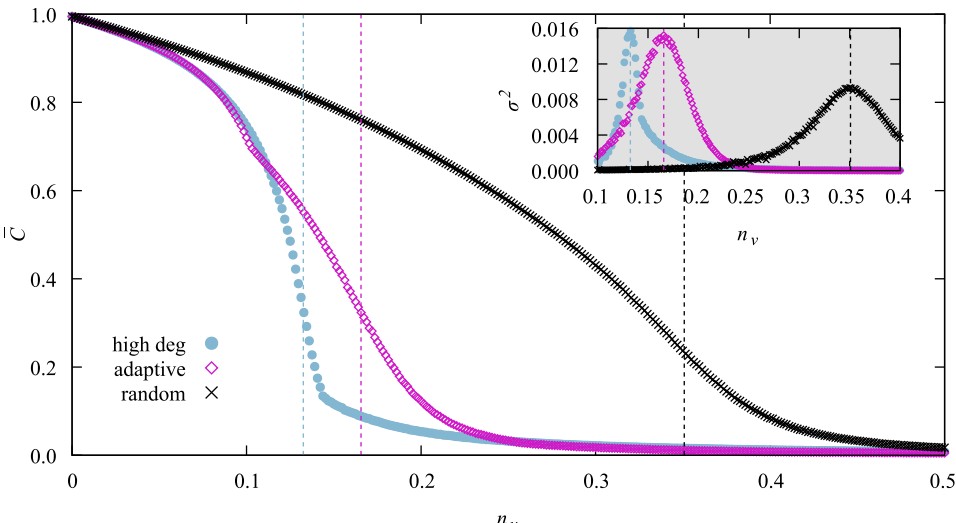

**Fig 1. Average cumulative fraction $\bar{C}$ of infected nodes for a typical network of $N = 3200$ nodes for different fractions $n_v$ of vaccinated nodes and the three applied vaccination strategies.** The inset shows the variance $\sigma^2(C)$. The data is obtained by averaging over 10000 samples per point and error bars are not shown as they are smaller than symbol size. The dashed lines indicate the positions of the peaks of the variance, respectively.

To understand why the adaptive strategy is performing worse than the non-adaptive one we initialized the network as if we wanted to perform an outbreak simulation. We then removed all vaccinated nodes and calculate the size of the remaining connected components and summed up the size of the components that contained any infected nodes, as this size represents an upper limit for the number of nodes that the disease can reach. We denote this quantity with $S$. The results are displayed in Fig 2, where we also include an upper bound which is given by the total number of remaining non-vaccinated nodes.

In panel (a) of the figure we see the results for the actual value $p = 0.1$ of the rewiring probability used throughout the paper. Clearly the size $S$ decreases more rapidly for the non-adaptive high degree heuristics compared to the adaptive one.

By looking at (b) we can understand why. A rewiring probability of $p = 0$ means that we never rewire and thus retain the original ring structure the network is initiated with. Here every node has degree 4, which means that the non-adaptive high degree heuristics and the random vaccination heuristics are equivalent. For this reason, these two cannot be distinguished in the figure. The adaptive high degree heuristics however will, due to its adaptive nature, distribute the vaccinations more or less evenly over the ring, as nodes whose neighbors have been vaccinated have a reduced degree. This makes splits of components less likely. This leads to $S(n_v)$ following the upper bound for quite some range of values of $n_v$ before the network finally splits up into more components.

In contrast if we look at (c), where we chose a rewiring probability of $p = 1$, which results in random networks similar to Erdőes-Rényi [72] networks with the difference being that due to the construction each node has at least degree 2, we see a different picture. The underlying ring structure is destroyed. Therefore the adaptive strategy performs better leading to reduced values of $S$ as compared with the other heuristics.

Note that in all three cases for any value of $n_v$ by using adaptive high-degree heuristics more edges are effectively removed than when using the other two strategies (not shown), which corresponds to the intuitive expectation. As we have seen, it is only the special small-world

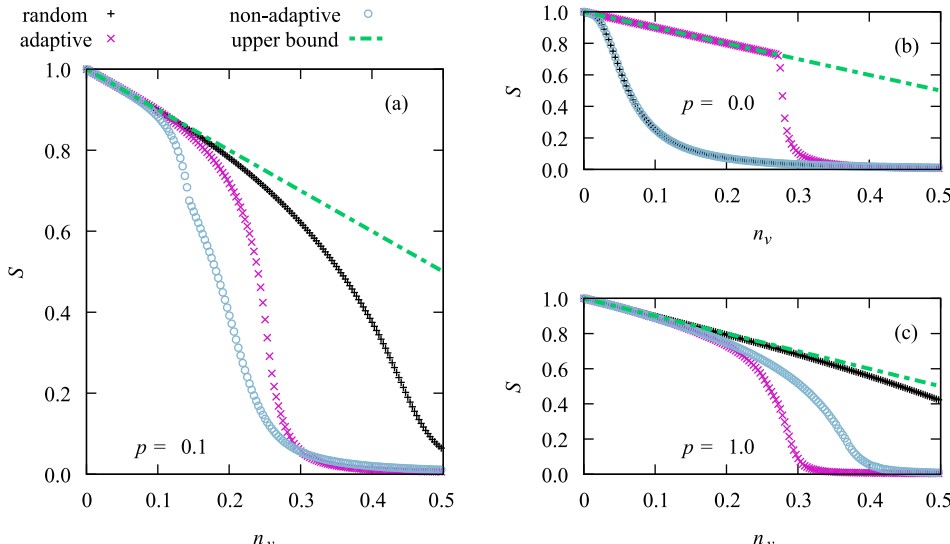

**Fig 2. Sum $S$ of relatives sizes of the connected components that contain the initial infections, shown as function of $n_v$.** Results are for a network of size $N$ = 3200, for the different vaccination strategies. Panels (a), (b) and (c) contain the results for different rewiring probabilities $p \in \{0, 0.1, 1\}$. Each data point was averaged over 20000 samples. Error bars are smaller than the symbol size. We also include an upper bound for the cluster size, which is the fraction of the network that was not yet vaccinated.

network structure which makes the adaptive vaccination strategy perform in some range of values for $n_v$ worse than the non-adaptive high-degree strategy. A similar effect, i.e., that the average inverse geodesic length, which can be used to quantify the severity of vertex removal in other contexts, of the adaptive high-degree heuristics can perform worse than the one of the non-adaptive heuristics, was observed before [71].

## Results: Large deviation sampling

By using the large-deviation approach we can determine the probability density function of interest, i.e., $P(C)$, over a large range of or even on its full support.

In Fig 3 we show the probability density functions for the random heuristics for different system sizes $N$ measured at their respective critical vaccination fraction $n_v^c$.

The pdf exhibits two peaks, one near $C \approx 0$ one around $C \approx 0.25$. Both peaks move towards smaller values for increasing system size and become sharper. We can also see that the data gathered by large-deviation sampling agrees very well with the simple sampling results—at least in the range where simple sampling was able to reach.

From the pdfs we can calculate the empirical *rate functions*

$$\Phi(C, N) \coloneqq -\frac{\ln P_N(C)}{N} + \Phi_0^N \tag{3}$$

where $\Phi_0^N$ is a constant that shifts the resulting rate function such that they all have their respective minimum at $\Phi_{\min} \equiv \min_C \Phi(C, N) = 0$. The idea behind calculating the rate function is to verify whether the probability density functions obey the so called *large-deviation principle* as described by large-deviation theory [35, 37–39]. If it obeys this principle the shape of the pdf has to follow the standard finite-size behavior

$$P_N(C) \sim e^{-N\Phi(C,N) + o(N)} . \tag{4}$$

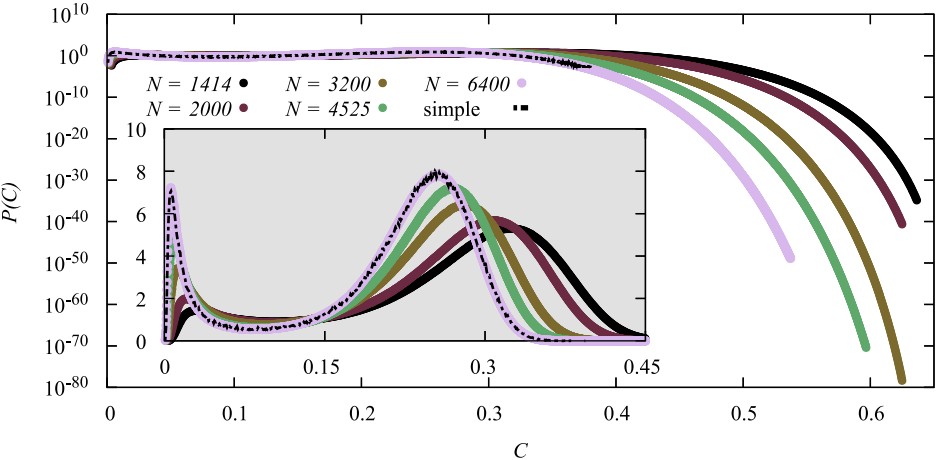

**Fig 3. Probability density of the cumulative number *C* of infections for different system sizes *N* at their critical vaccination doses $n_v^c$ with the random vaccination strategy as measured by the large-deviation algorithm.** It also includes simple sampling data for the largest system size. Linear scale in the inset.

Note that the size *N* is primarily important as a linear prefactor to the rate function, while other dependencies on the size are less important, as denoted by *o*(*N*), i.e., their contributions vanish relatively in the limit $N \to \infty$. From Eq (4) it is also clear why at the minimum $\Phi_{\min} = 0$ should hold, because if $\Phi_{\min} > 0$ the probability would converge to zero everywhere while if $\Phi_{\min} < 0$, *P*(*C*) would diverge for some values of *C*. To check the large deviation principle is of interest, because models for which the principle holds are better accessible by analytical approaches, e.g., by using the Gärtner-Ellis theorem [35, 37–39].

The rate functions we calculated from the pdfs of Fig 3 are shown in Fig 4. In the range $C \in$ [0, 0.25] it looks like the functions converge to a limiting shape for $N \to \infty$. Right to the second minimum of the rate function, which corresponds to the maximum of the pdf at around $C \approx$

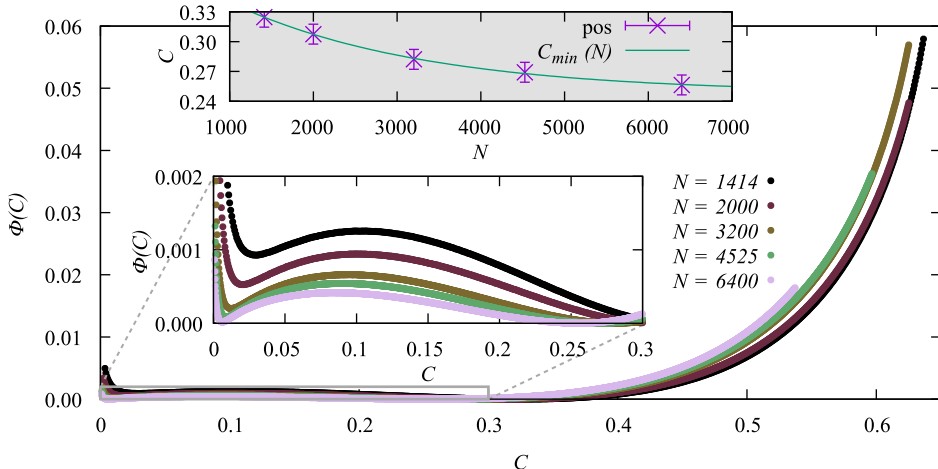

**Fig 4. Rate functions Φ(*C*) for the random vaccination heuristics for different system sizes *N* at their respective critical vaccination.** The gray inset shows the position of the minimum of the rate function as a function of *N* as well as the fit fit-exp with *a* = 0.142(5), *b* = 0.00044(4) and $C_{\min}^\infty = 0.248(3)$. The other inset shows a zoom for better visibility.

0.25, the case is less clear and some of the functions for different system sizes even overlap. If we shift each function horizontally such that their minimum appears at the same value of $C$ (not shown) the functions exhibit a monotonous $N$-dependence also in the region of large values of $C$, indicating a convergence as well.

The position of the minimum is also shown in Fig 4 and seems to follow an exponential function. We therefore fitted the function

$$C_{\min}(N) = a \; e^{-b\,N} + C_{\min}^{\infty} \,, \tag{5}$$

which worked pretty well.

Additional pdfs and rate functions for the other heuristics at their respective critical points look very similar and are thus moved to S1 Appendix.

In Fig 5 we show the comparison of the different vaccination strategies at their respective critical vaccination value. The probability density functions for both high-degree heuristics are almost identical even though the adaptive high-degree heuristics vaccinated about 3% more nodes than the non-adaptive one. The shape for the random heuristics is a bit different. The peaks of $P(C)$ for this vaccination heuristics are shifted to the left and appear sharper as compared to the two high-degree heuristics. Also the probabilities for large values of $C$ are significantly smaller and the decline in probability is steeper.

In Fig 6 we compare all three heuristics with the same vaccination doses $N_v = 530$, which is the critical vaccination value for the adaptive high-degree heuristics for the system size $N = 3200$. Looking at the different pdfs we can see three different general shapes. The pdf of the non-adaptive high degree heuristics exhibits one maximum at $C \approx 0.05$ and strongly declines afterwards. The pdf for the adaptive high degree heuristics exhibits two maxima that are comparable in height (better visible in S1 Appendix) and a sharp decline in probability after the second one. Lastly the random heuristics also exhibits two maxima, however the second maximum is at a very large $C$ value and the first peak is at noticeably lower probabilities. The probability also declines below $10^{-20}$ in between both maxima, i.e., they are much stronger separated.

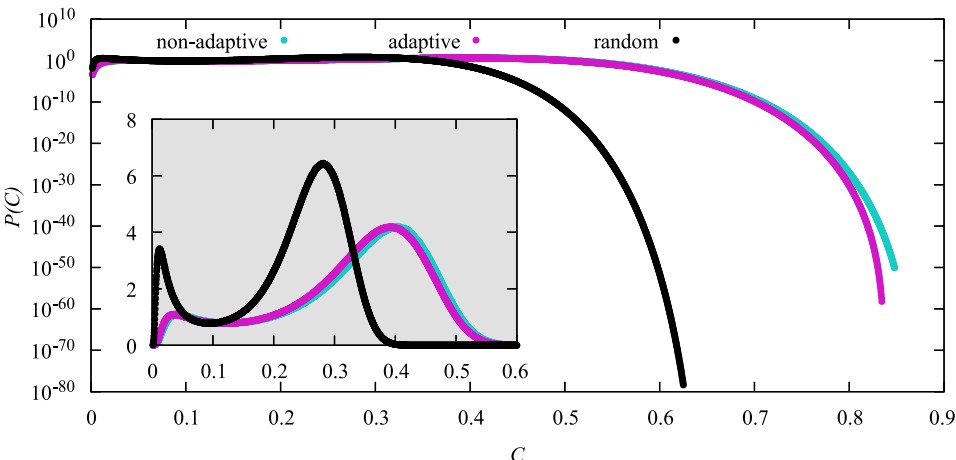

**Fig 5. Probability density functions of the cumulative fraction $C$ of infections for $N = 3200$ for the different vaccination heuristics at their respective critical point.** For the adaptive high-degree heuristics that corresponds to $N_v = 530$, for the non-adaptive one to $N_v = 424$, and for the random heuristics to $N_v = 1133$. Linear scale in the inset.

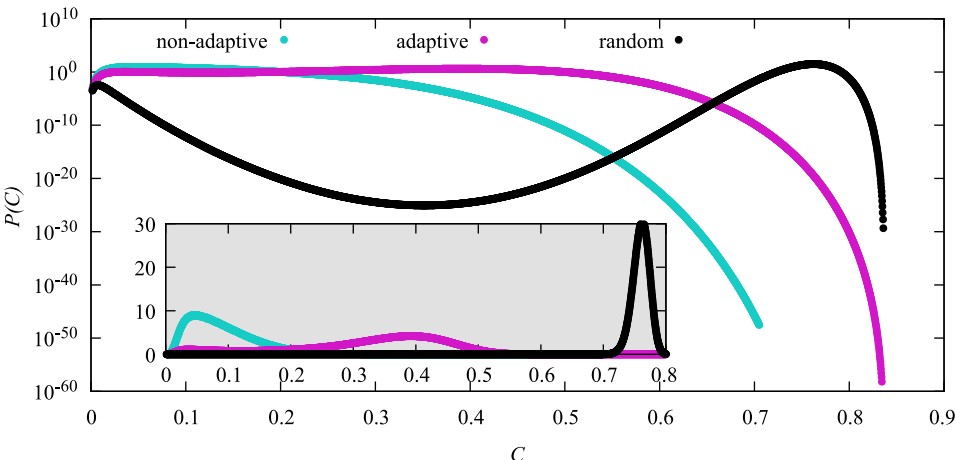

**Fig 6. Probability density functions of the cumulative fraction $C$ of infections for $N = 3200$ and $N_v = 530$ for the different vaccination strategies.** Linear scale in inset.

Overall we can see that the non-adaptive high-degree heuristics proved very effective, followed by the adaptive high degree heuristics whereas the random heuristics is the least-efficient of them.

Now we want to see how the distributions look like when we choose values of $N_v$ such that the distributions exhibit a maximum roughly at the same values of $C$, i.e., where the non-adaptive high degree heuristics exhibits its maximum in Fig 6. This allows us to compare the mere shape of the distributions.

The achieve this, we first performed a few simple sampling simulations for various values of $N_v$ to align the $C$ positions of the maxima before running the large-deviation simulations. It turned out that it is not possible to find a value of $N_v$ to align the random heuristics well, because here the peak position is too insensitive to the value of $N_v$, as demonstrated by Fig 7.

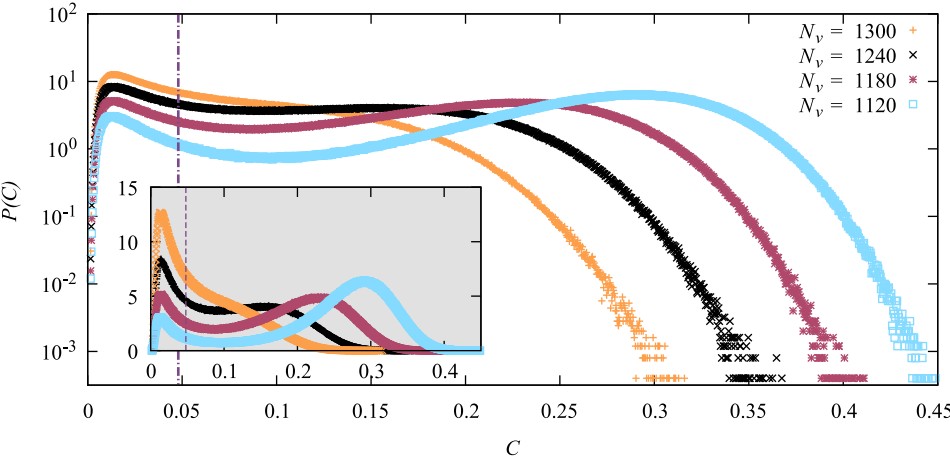

**Fig 7. Probability density functions of the cumulative fraction $C$ of infections for the random heuristics, $N = 3200$, and for different number $N_v$ of vaccinated nodes.** The histograms were measured via simple sampling using 8000000 samples each. Linear scale in the inset. The dashed line indicates the position of the maximum that we aimed for.

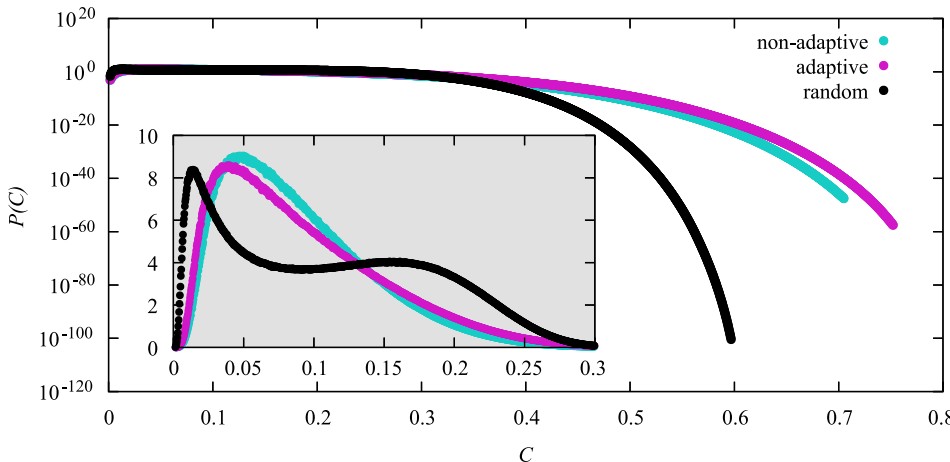

**Fig 8. Probability density functions of the cumulative fraction $C$ of infected nodes for $N = 3200$ for the different vaccination strategies.** The non-adaptive high-degree heuristics used $N_v = 530$, the adaptive one used $N_v = 670$ and the random heuristics used $N_v = 1240$. Linear scale in inset.

Thus we had to compromise for the random heuristics. The measured pdfs are displayed in Fig 8.

Both high-degree heuristics display a very similar behavior, whereas the random heuristics displays two peaks in the range of small values of $C$ and exhibits a much steeper decline in probability for increasing values of $C$ resulting in much smaller probabilities for the tail. Thus, while exhibiting about the same typical number of infections, strong fluctuations are much more reduced for the random heuristics, at the price of a much larger number of vaccinated nodes.

With the same idea in mind we now wanted to align the maxima to the position of the maximum for the random heuristics from Fig 6. Again, we used simple sampling to align the maxima and did large deviations afterwards. This time no issue occurred during the alignment and the results of the large-deviation simulations can be found in Fig 9.

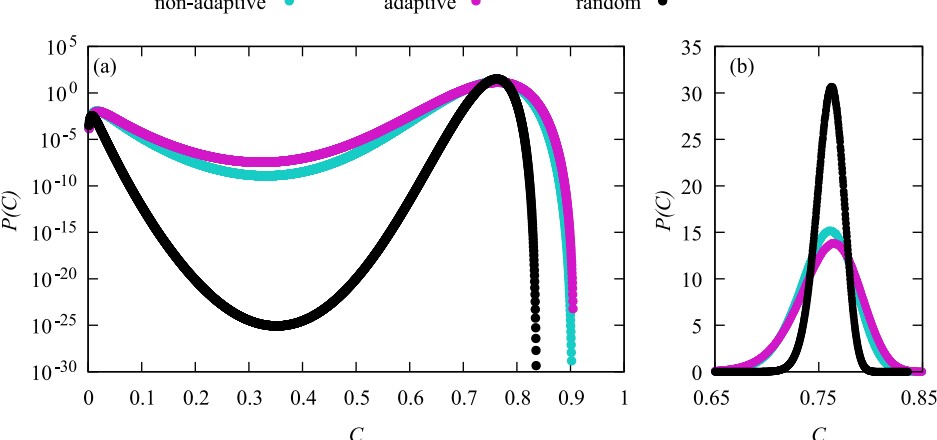

**Fig 9. Probability density functions of the cumulative fraction $C$ of infections for $N = 3200$ for the different vaccination strategies.** The non-adaptive high-degree heuristics used $N_v = 315$, the adaptive one used $N_v = 305$ and the random heuristics used $N_v = 530$. On the left we show the data in logarithmic scale and on the right the range $C \in [0.65, 0.85]$ in linear scale.

Again both high-degree heuristics give very similar results. The general shape of the pdf for the random heuristics is also similar, and all curves exhibit two maxima at roughly the same positions and with the same magnitudes. Apart from that, for the random heuristics the probability to observe intermediate values $C$ is much lower as compared to the other two heuristics, i.e., the fluctuations away from typical values are again much more reduced.

## Infection course analysis

In previous work on the SIR dynamics without protections measure [36] we have observed that the infection dynamics are remarkably different for different regions of the distributions $P(C)$. This allowed us to pin-point characteristic properties of mild, typical and severe outbreaks. This might help to design or improve suitable protection measures. To investigate whether such characteristic properties can be identified in the presence of vaccinations, we stored, during the entropic sampling, every $X$'th disease time series, where we calculated X in such a way that we store 2000000 time series in total per entropic sampling simulation. These disease time series were binned according to their respective value of $C$.

**Disparity.**   To compare the time series with one another we now computed their *disparity* [36], which is defined as follows. Let us first focus on two arbitrary time series $X_1, X_2$, where $X(\tau)$ corresponds to $c(\tau)$ or $i(\tau)$ with the only difference that we normalize them by dividing every entry $X(\tau)$ by the maximum, i.e., $\max_\tau X(\tau)$, with the intention of only investigating the shape and not the magnitude of the curves. Note that the time series $X_1$ and $X_2$ may have different lengths $l_1, l_2$ and we thus define $l_{max} = \max\{l_1, l_2\}$. If necessary we continue a time series beyond its given length by repeating its last value, which makes sense because for larger time usually no further changes occur. In this way both series have $l_{max}$ entries.

This lets us define the *distance*

$$d(X_1, X_2) = l_{max}^{-1} \sum_{\tau=0}^{l_{max}} |X_1(\tau) - X_2(\tau)| \,. \tag{6}$$

We now define the disparity $V_X(C_1, C_2)$ for two different values of $C$ as averaged distance $d$. The average is performed over all pairs of time series where time series $X_1$ exhibits the bin value $C = C_1$ and the other $C = C_2$, respectively.

In Fig 10 we show the disparity heat maps for the three vaccination heuristics as calculated from the normalized $i(\tau)$ time series. The pdfs $P(C)$ that correspond to these disparities were already displayed in Fig 6. Note that the diagonals represent the comparisons within a single bin, i.e., of time series that lead to the same cumulative number of infections $C$. Thus, they represent the amount of variation of the time series for a given value of $C$.

Looking at the heat map for the non-adaptive high degree heuristics we make a few observations. Along the diagonal, the disparity increases with growing value of $C$. Thus two different disease-evolutions which exhibit the same value of $C$ are more similar if $C$ is small. Apart from that one can see a cone-shape of the boundaries between similar values of $V$.

The heat map for the adaptive strategy in Fig 10 looks rather similar. Though this time, along the diagonal, the time series are becoming more dissimilar until $C \approx 0.3$ and become more similar again if $C$ is increased further, though this effect is very small and almost not visible.

The heat map for the random heuristics, displayed on the right in Fig 10 looks different. Here we can see three regions, the first, $C < 0.1$, is a region of curves that are similar to one another. The second, $0.1 \le C < 0.35$, consists of a slightly higher disparity and the last region, $0.35 \le C$, consists of curves that are very similar to one another. Thus, also for this quantity of measurement, which tells more about the details of the disease evolution, the random

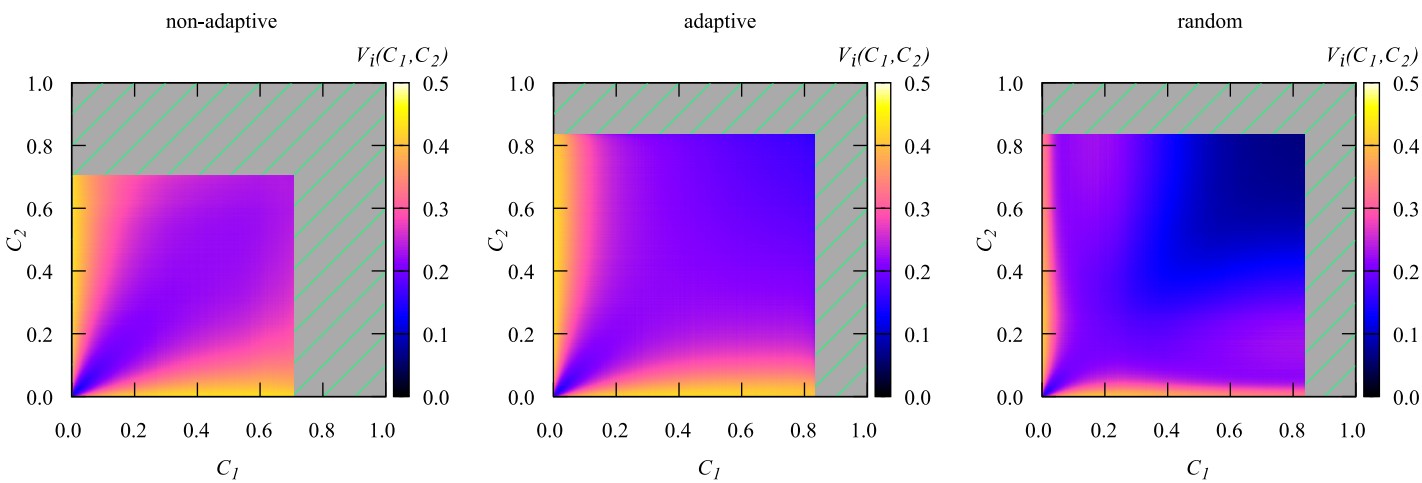

**Fig 10. Color-coded disparity $V_i(C_1, C_2)$ of the $i(\tau)$ time series for $N = 3200$, $N_v = 530$.** On the left we show the results for the non-adaptive high-degree vaccination, in the middle the results for the adaptive high-degree vaccination and on the right the results for the random vaccination. The time series are binned with their corresponding value of $C$ and for each bin 1500 randomly drawn time series are used for the calculation of the disparity $V_i(C_1, C_2)$. For the shaded area no time-series exist as it was outside of the interval used for WL.

heuristics leads to smaller fluctuations as compared to the degree-based ones. Interestingly, when fixing $C_1$ at a small value like 0.05 and varying $C_2$, one observes that first the disparity increases, but at much larger values of $C_2$ the shape of the time evolution become more similar again. This resembles "reentrant" behavior sometimes observed for phase transitions.

In Fig 11 we show some examples of the time-series used to create the heat maps to better understand the behavior we observed when analyzing the disparities.

On the left we see the curves for the non-adaptive high degree heuristics. For small values of $C$, $i(\tau)$ peaks close to the beginning before the disease quickly goes extinct. For larger values of $C$ the time-series display a strongly fluctuating behavior, where no clear peak exists and the disease takes way longer to go extinct, though the actual duration varies a lot. Note that strong variations make the evolution of diseases very hard to predict. On average, as the value of $C$ increases, the duration of the outbreak and the maximum number of simultaneously infected

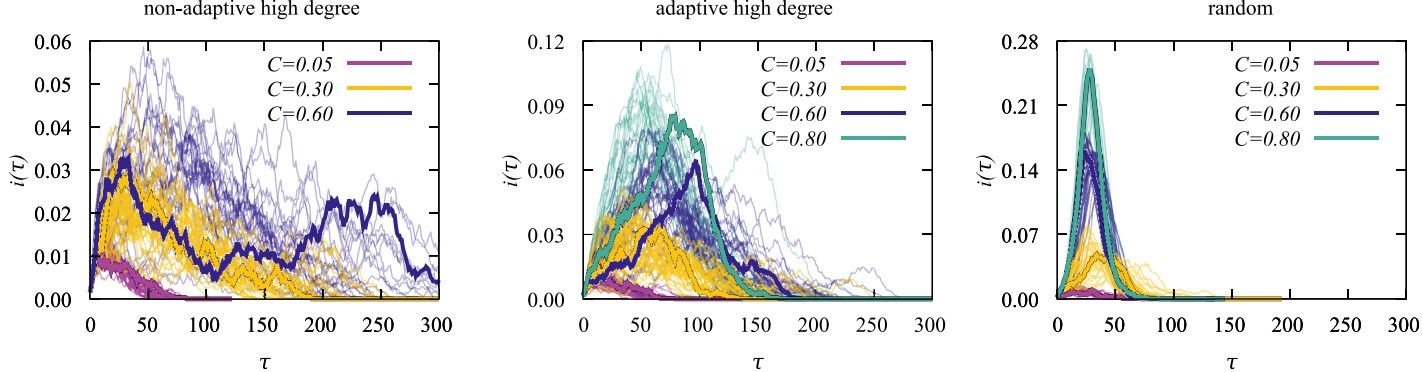

**Fig 11. Infection time series $i(\tau)$ for different values of $C$.** These time series were created during the entropic sampling with a Network of size $N = 3200$ and a vaccination doses of $N_v = 530$. On the left we show the results for the non-adaptive high-degree heuristics, in the middle for the adaptive high-degree heuristics and on the right for the random heuristics. For each value of $C$ we plot 30 time series. We also included one arbitrary time series for each $C$ value that we highlighted for clarity.

nodes also increase. However, the maximum number of affected nodes is relatively small, with at most 6% of the network affected at once

In the center of Fig 11 we present the time series corresponding to the adaptive heuristics. Compared to the plot on the left we see a tendency for the curves to peak earlier, i.e., within the first 100 time steps, and the disease dies out quicker overall. Also some remnants of the strongly fluctuating behavior of the left plot can be seen. However, for the adaptive heuristics, the most likely trajectories are associated with higher values of C in comparison to the non-adaptive heuristics meaning that the fluctuations themselves correspond to more probable outbreak scenarios.

The time-series corresponding to the random heuristics are displayed in Fig 11 on the right. We see that the time series for large $C$ display a clear peak and Gaussian-like shape. Still, the time series for values of $C$ near $C = 0.3$ are still a bit more fluctuating and correspond to slowly-developing diseases as were observed also for the case without vaccinations [36]. These outbreaks tend to take the longest to become extinct. Overall the heat map and time series that are seen for the random case here look very similar to the ones without vaccinations [36]. Obviously, random vaccination alters the structure of the network of non-vaccinated nodes not very strongly.

**Conditional densities.**   For the health care system it is important to estimate the total number of people that will be infected at once or rather the maximum of this quantity, because this should not exceed the capacity of the healthcare resources. We therefore look at the maximum

$$M := N \max_{\tau} i(\tau) \ . \tag{7}$$

To investigate how $M$ correlates with the cumulative number of infections we calculate the conditional density $\rho(M|C)$, i.e., the probability for a disease outbreak to exhibit a specific value of $M$ given the value for $C$. In Fig 12 we show this conditional density for $N = 3200$ and $N_v = 530$, i.e., calculated during the same simulation we used to generate Fig 6, such that we can use that figure to know how probable a given range of $C$ values is. Note that most parts of $\rho(M|C)$ are for values of $C$ where $P(C)$ is extremely small, i.e. they are accessible only by using the large-deviation approach.

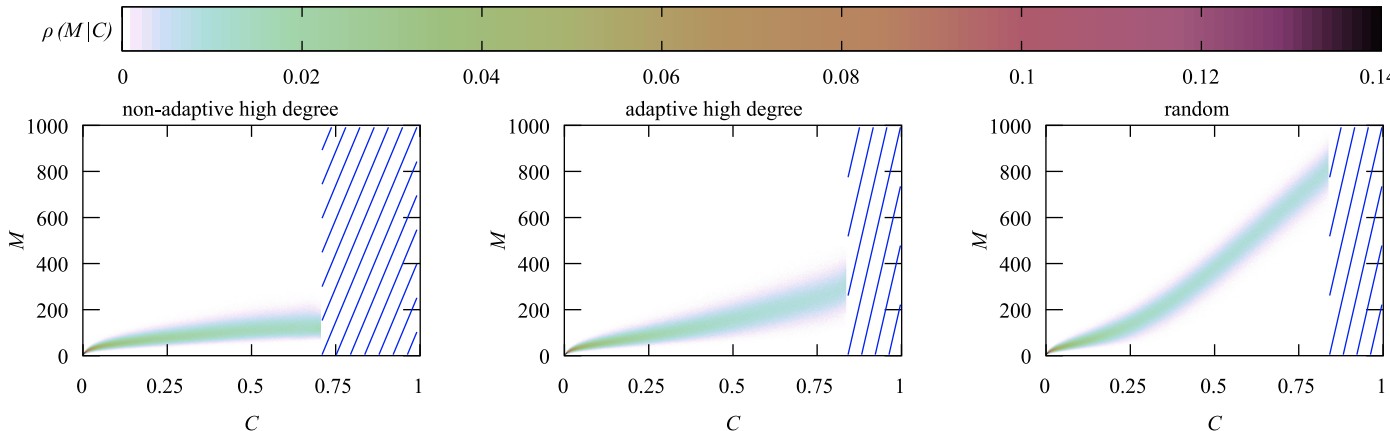

**Fig 12. Color-coded conditional densities $\rho(M|C)$ for a network of $N = 3200$ nodes and $N_v = 530$ vaccination doses for the three analyzed vaccination strategies.** The blue stripes indicate the $C$ range that was outside of the Wang-Landau interval.

For the non-adaptive high degree heuristics the system is beyond its critical vaccination value and $C < 0.1$ is highly likely. Since the disease dies out quickly in this range, it results in very small values for $M$. Only for very small values of $C$, the maximum $M$ increases strongly. Even if we look at the less likely outcomes, $M$ always stays below 200.

The density for the adaptive heuristics looks up to $C \approx 0.4$ very similar to the non-adaptive one. For larger values of $C$, outbreaks tend to lead to higher values of $M$, though overall $M$ stays below 400. When considering the pdf in Fig 6, we can conclude that $M < 200$ is the most likely outcome here and the large values of $M$ are not so relevant when deciding about the capacities of health-care systems.

For the random heuristics up until $C < 0.25$ its conditional density is very similar to the other two heuristics. Then, however, $M$ almost linearly increases with $C$ and the incline is much steeper than seen for the high-degree heuristics. For the random heuristics $C > 0.7$ is the most probable range and thus a typical outbreak exhibits $M > 600$, leading to a much higher load on the health-care system.

Next, we wanted to investigate the speed with which the disease propagates through the network. One could investigate the entire duration of the disease, but it makes more sense to go about it differently. In the real-world it is unlikely that the disease will get detected immediately. We want a quantity to reflect that. We use the $c(\tau)$ time series and measure the number of time steps $\tau_1$ it takes until $c(\tau_1) = 0.1 \times c(\infty)$ and the number of time steps $\tau_2$ it takes until $c(\tau_2) = 0.9 \times c(\infty)$ is reached. The we define $\tau_{10}^{90} = \tau_2 - \tau_1$, i.e., we measure the number of time steps the disease required to reach 90% of its final cumulative number of infections once it already reached 10%, for each run.

We calculated the conditional densities $\rho(\tau_{10}^{90}|C)$ for the different vaccination strategies for $N = 3200$ and $N_v = 530$ and show the results in Fig 13. Again we can use Fig 6 to see how probable each value of $C$ is.

All strategies exhibit a similar incline for $C \leq 0.05$. The disease durations are rather short and do not fluctuate much. This can be seen as one specific phase within range of possible $C$ values.

On the other hand, for the non-adaptive strategy we see a large spread of duration times for $C > 0.05$ and this spread increases for increasing value of $C$. This means it will be hard to predict the duration of this disease even if the value of $C$ that will be reached in total was known.

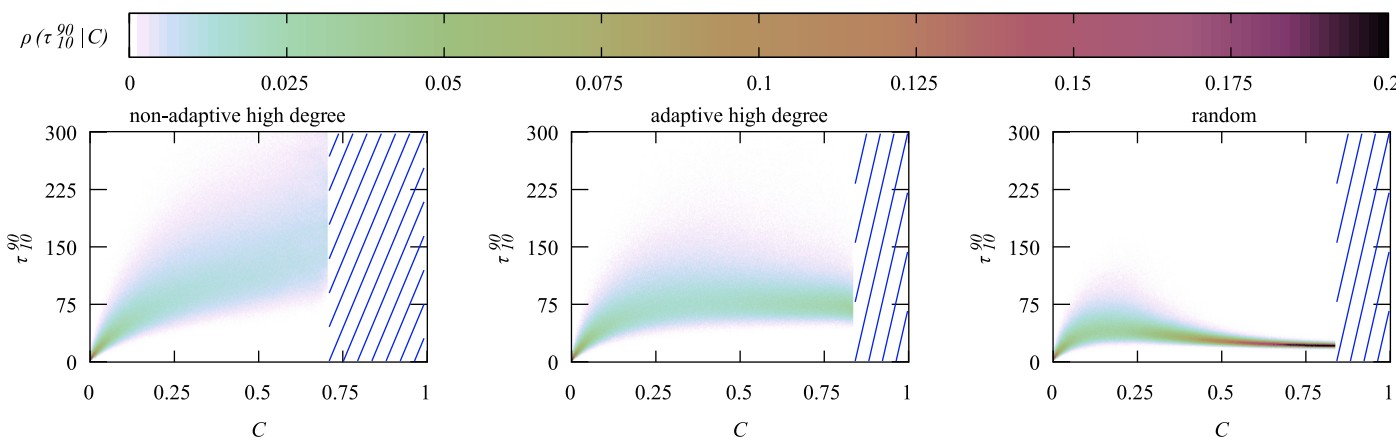

**Fig 13. Conditional density $\rho(\tau_{10}^{90}|C)$ that shows the probability of $\tau_{10}^{90}$ for any given value of $C$ for $N = 3200$ and $N_v = 530$ for the different vaccination strategies.** The blue stripes indicate the $C$ range that was outside of the Wang-Landau interval.

This occurrence of a strong-fluctuation phase fits with the observations from Fig 11 where we saw a fluctuation-rich behavior for the corresponding time series.

The adaptive strategy shows a similar behavior for $C > 0.05$, though the durations tend to be a bit lower and the spread does not become as large as for the non-adaptive case. For $C > 0.5$ the durations tend to become a tiny bit shorter again and the spread of duration times also decreases, but only slightly. This correspond to the slightly higher values of $M$, i.e., the diseases are more concentrated in time.

The spread of durations for the random heuristics is much smaller overall and peaks at around $C \approx 0.2$, which is in a rather improbable range. For $C > 0.5$ the spread is almost non-existent and $\tau_{10}^{90}$ becomes constant, which is also reflected in Fig 11 where we can see a very regular behavior. This can be seen as the presence of a third phase for large values of $C$, in contrast to the two degree-based heuristics.

Thus, for the random heuristics the disease outbreaks are stronger, but can be predicted better due to smaller fluctuations.

## Summary and outlook

We investigated the effect vaccinations on the dynamics of disease spreading for the standard SIR model. We considered three different vaccination strategies, namely a random vaccination heuristics and two different heuristics which target high-degree nodes first.

For this purpose we chose a rather large transmission probability of $\lambda = 0.4$ such that the model would, without vaccination, exhibit a strong epidemic outbreak.

Using large-deviation methods and especially the $1/t$ Wang-Landau algorithm [63] we were able to calculate the probability distribution $P(C)$ of the cumulative fraction $C$ of infected nodes over a large range of its support and down to probabilities as small as $10^{-80}$. While probability densities such as $10^{-10}$, which are out of range of standard approaches, might be even useful for practical considerations, obtaining $P(C)$ over a large range of the support is in particular satisfying from a scientific point of view, because we are able to obtain a quantity which is analytically not accessible.

Note that some results exist for chemical reaction networks [28]. Since they are treated typically on mean-field level, the corresponding distribution functions usually have a simple structure, like a multidimensional simple Gaussian. Still, there are also known cases where distributions with multiple peaks have been found, like for a simple four-species reaction network [29]. Therefore the multi-modal structure of the distributions we observe might not be due to the complex underlying network, although all details of $P(C)$ will certainly depend on the network structure.

In particular we investigated the empirical rate functions and found that the convergence properties were less clear than in the pure SIR case [36]. Nevertheless, when considering that the finite-size dependence includes horizontal shifts on the $C$ axis as well as changes on the probability axis, it appears likely that the large-deviation principle is fulfilled. Thus it appears likely, that analytical calculations of the probability densities are possible with, e.g., the Gärtner-Ellis theorem.

By additionally applying an entropic sampling algorithm [69], we were able to investigate the actual time-series corresponding to different ranges of the value of $C$. Having available these sample time series of typical as well as extreme behavior might allow one to understand strong or weak outbreaks better which could lead to more sophisticated strategies.

We compared the time-series of the three vaccination strategies given the same network and same vaccination amount. We observed that for most of the cases with targeted high-degree vaccination a behavior with strong fluctuations dominates. This effect itself is less

pronounced for the adaptive heuristics. For the random heuristics, in principle three phases can be observed along the $C$ axis. In general, fluctuations are mostly suppressed, making disease propagation better predictable, although the strategy is less efficient as compared to the degree-based ones.

For practical applications, one could draw from these results the strategy to use a targeted high-degree heuristics to vaccinate enough nodes to exclude a pandemic, but to additionally vaccinate a certain fraction of randomly chosen nodes, to reduce the fluctuations such that the infection dynamics becomes more regular and predictable. This would probably also increase the acceptance of vaccination measures in the society.

The approaches used here is very general, such that in can be applied for many different disease-spreading problems, such as those exhibiting different infected states, including the implementation of other counter measures, or processes which take place on dynamic networks.

Furthermore, the large-deviation approach could be even more useful, if one considers coupled networks of multiple species. This would allow one to investigate the transfer of zoonoses between animals and humans. In this way one could study the fundamental processes or conditions that are required to let a disease transfer between different host species and possibly result in a huge outbreak in the new host species. Note that for relevant outbreaks these transfer probabilities are necessarily small, because for the cases with a high transfer probability, they will have happened during evolution already. Thus, a large-deviation approach is very suitable in this case.

## Supporting information

**S1 Appendix.**
(PDF)

**S1 Data.**
(GZ)

**S1 Fig.**
(EPS)

**S2 Fig.**
(EPS)

**S3 Fig.**
(EPS)

**S4 Fig.**
(EPS)

## Author Contributions

**Conceptualization:** Yannick Feld, Alexander K. Hartmann.

**Funding acquisition:** Yannick Feld.

**Investigation:** Yannick Feld.

**Methodology:** Yannick Feld, Alexander K. Hartmann.

**Project administration:** Alexander K. Hartmann.

**Software:** Yannick Feld.

**Supervision:** Alexander K. Hartmann.

**Validation:** Yannick Feld, Alexander K. Hartmann.

**Visualization:** Yannick Feld.

**Writing – original draft:** Yannick Feld.

**Writing – review & editing:** Yannick Feld, Alexander K. Hartmann.

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
