## [Decision Letter · Decision Letter 0]

8 Jun 2023

PONE-D-23-12906Large-deviations of disease spreading dynamics with vaccinationPLOS ONE

Dear Dr. Feld,

Thank you for submitting your manuscript to PLOS ONE. After careful consideration, we feel that it has merit but does not fully meet PLOS ONE’s publication criteria as it currently stands. Therefore, we invite you to submit a revised version of the manuscript that addresses the points raised during the review process.

We look forward to receiving your revised manuscript.

Kind regards,

Tiago Pereira

Academic Editor

PLOS ONE

Reviewers' comments:

Reviewer's Responses to Questions

**Comments to the Author**

1. Is the manuscript technically sound, and do the data support the conclusions?

Reviewer #1: Yes

2. Has the statistical analysis been performed appropriately and rigorously? 

Reviewer #1: Yes

3. Have the authors made all data underlying the findings in their manuscript fully available?

Reviewer #1: Yes

4. Is the manuscript presented in an intelligible fashion and written in standard English?

Reviewer #1: Yes

5. Review Comments to the Author

Reviewer #1: I thoroughly enjoyed reading this paper. The usage of Wang-Landau Algorithm for epidemic models introduces a refreshing novelty. However, I would like to address a minor concern regarding the lack of discussion on how the obtained results correlate with previous theoretical findings.

In the realm of studying Markov counting processes, there exists a substantial body of literature [1] exploring the relationships between these models and the associated differential equations. Particularly, deviation results have been extensively examined [2]. I would appreciate if the authors could provide a brief analysis, perhaps just a few lines, comparing their experimental results with these theoretical findings. This discussion would be instrumental in determining whether the imposed graph structure disrupts the classical results, as they typically assume a fully connected graph.

References:

[1] Anderson, D. F., & Kurtz, T. G. (2011). Continuous time Markov chain models for chemical reaction networks. In Design and analysis of biomolecular circuits: engineering approaches to systems and synthetic biology (pp. 3-42). New York, NY: Springer New York.

[2] Bibbona, Enrico, and Roberta Sirovich. "Strong approximation of density dependent Markov chains on bounded domains." arXiv preprint arXiv:1704.07481 (2017).

6. PLOS authors have the option to publish the peer review history of their article (what does this mean?). If published, this will include your full peer review and any attached files.

Reviewer #1: No

---

## [Author Response · Author response to Decision Letter 0]

13 Jun 2023

See attached "Cover letter" and response to reviewer

---

## [Editor Report · Decision Letter 1]

15 Jun 2023

Large-deviations of disease spreading dynamics with vaccination

PONE-D-23-12906R1

Dear Dr. Yannick Feld,

We’re pleased to inform you that your manuscript has been judged scientifically suitable for publication and will be formally accepted for publication once it meets all outstanding technical requirements.

Kind regards,

Tiago Pereira

Academic Editor

PLOS ONE

Additional Editor Comments (optional):

The authors address the referee's comments accordingly. Since the comments were minor we didn't resend the paper to the referee. 
---

## [Editor Report · Acceptance letter]

29 Jun 2023

PONE-D-23-12906R1 

Large-deviations of disease spreading dynamics with vaccination 

Dear Dr. Feld:

I'm pleased to inform you that your manuscript has been deemed suitable for publication in PLOS ONE. Congratulations! Your manuscript is now with our production department. 

Kind regards, 

on behalf of

Dr. Tiago Pereira 

Academic Editor

PLOS ONE